# Investigation of Biomaterial Ink Viscosity Properties and Optimization of the Printing Process Based on Pattern Path Planning

**DOI:** 10.3390/bioengineering10121358

**Published:** 2023-11-26

**Authors:** Jiahao Wu, Chunya Wu, Siyang Zou, Xiguang Li, Bo Ho, Ruijiang Sun, Chang Liu, Mingjun Chen

**Affiliations:** 1State Key Laboratory of Robotics and System, Harbin Institute of Technology, Harbin 150080, China; wjh1079946368@163.com (J.W.); zouzsy@hotmail.com (S.Z.); lixiguang1996@163.com (X.L.); houbo275134081@163.com (B.H.); srjedu2019@163.com (R.S.); mihawk_lc@163.com (C.L.); chenmj@hit.edu.cn (M.C.); 2Key Laboratory of Micro-Systems and Micro-Structures Manufacturing, Harbin Institute of Technology, Harbin 150080, China

**Keywords:** bioprinting, response surface methodology, power law, printability, simulation, print path

## Abstract

Extruded bioprinting is widely used for the biomanufacturing of personalized, complex tissue structures, which requires biomaterial inks with a certain viscosity to enable printing. However, there is still a lack of discussion on the controllable preparation and printability of biomaterial inks with different viscosities. In this paper, biomaterial inks composed of gelatin, sodium alginate, and methylcellulose were utablesed to investigate the feasibility of adjustment of rheological properties, thereby analyzing the effects of different rheological properties on the printing process. Based on the response surface methodology, the relationship between the material components and the rheological properties of biomaterial inks was discussed, followed by the prediction of the rheological properties of biomaterial inks. The prediction accuracies of the power-law index and consistency coefficient could reach 96% and 79%, respectively. The material group can be used to prepare biomaterial inks with different viscosity properties in a wide range. Latin hypercube sampling and computational fluid dynamics were used to analyze the effects of different rheological properties and extrusion pressure on the flow rate at the nozzle. The relationship between the rheological properties of the biomaterial ink and the flow rate was established, and the simulation results showed that the changes in the rheological properties of the biomaterial ink in the high-viscosity region resulted in slight fluctuations in the flow rate, implying that the printing process for high-viscosity biomaterial inks may have better versatility. In addition, based on the characteristics of biomaterial inks, the printing process was optimized from the planning of the print pattern to improve the location accuracy of the starting point, and the length accuracy of filaments can reach 99%. The effect of the overlap between the fill pattern and outer frame on the print quality was investigated to improve the surface quality of complex structures. Furthermore, low- and high-viscosity biomaterial inks were tested, and various printing protocols were discussed for improving printing efficiency or maintaining cell activity. This study provides feasible printing concepts for a wider range of biomaterials to meet the biological requirements of cell culture and tissue engineering.

## 1. Introduction

Bioprinting is a rapid additive manufacturing technology in which biomaterials are assembled in desired locations on demand through computer-aided design [1,2]. Three-dimensional biological scaffolds constructed by bioprinting can facilitate cell growth in an environment that mimics the human body more closely [3,4]. It is also possible to fabricate personalized and complex biological scaffolds rapidly for different tissues and organs through bioprinting, which plays an important role in research in the medical field [5]. Bioprinting processes can be mainly classified into inkjet-based printing [6,7], extrusion-based printing [8,9], and digital light processing-based bioprinting [10]. Among them, the extrusion-based printing process is fairly popular due to the simple principle and a wide range of printable biomaterials [11,12,13]. However, different types of biomaterial ink materials may have different rheological characteristics, especially shear thinning, which restricts the versatility of the printing process, leading to repeated exploration of the printing process parameters. Therefore, it is necessary to investigate the relationship between rheological properties and flow states in order to analyze the application potentials of biomaterial inks with different rheological properties, as well as the versatility of the corresponding printing processes.

At present, many scholars have carried out a series of bioprinting process research studies focusing on the fidelity of prints. Ning et al. investigated the process parameters of extrusion bioprinting and optimized the printing performance in a Carbopol suspension bath by regulating the material concentration [14]. To overcome the limited types of printable bioinks, Ouyang et al. added thermally reversible gelatin to bioinks to form an interpenetrating network, which improves the printability of different bioinks [15]. Ribeiro et al. developed a systematic protocol to evaluate fidelity based on the analysis of the relationship between the collapse and fusion stress of filaments [16]. Yi et al. conducted a systematic investigation into additional GelMA types and porogen types, with the effects of the porogen concentrations and molecular weights clarified to enable the preparation of different microporous structures [17]. Although these works have attempted to reveal the underlying mechanism at different stages of the entire printing process, the printability of biomaterial inks with arbitrary viscosity properties still lacks systematic investigation. The challenge of preparing biomaterial inks with defined viscosity properties and predicted printability still remains significant.

To achieve the controllable adjustment of the viscosity for biomaterial inks, it is necessary to develop a stable material group to coordinate two parameters in the shear rate–viscosity curve, i.e., the power-law index and the consistency coefficient. As a product of the partial hydrolysis of collagen, gelatin is a natural polypeptide polymer [18] with similar biocompatibility to collagen, which allows cells to attach, proliferate, and grow [19,20,21]. Gelatin can be dissolved in a warm water bath, and the viscosity of the solution increases gradually as the temperature decreases. When the temperature is lowered to a certain threshold, gelatin forms a strong physical network composed of hydrogen bonds [22,23]. Sodium alginate is a polysaccharide extracted from natural brown algae with biological inertness [24], and its aqueous solution is a viscous colloid. Since the sodium ions can exchange with calcium, copper, zinc, and other divalent metal cations, producing the G-unit of sodium alginate that forms an “egg lattice” structure, sodium alginate is ready to be crosslinked to form a gel with good mechanical properties [24,25]. Methylcellulose is a product of full or partial methylation of the three hydroxyl groups of glucose in cellulose [26,27], with the solubility changeable by controlling the degree of substitution. Normally, methylcellulose dissolves in cold water, becoming a viscous liquid that can be mixed with other polymers to form biomaterial inks with good printability [28]. Since all three materials are natural polymers with a certain biocompatibility but different biochemical properties, they are often used in the bioprinting of blood vessels, scaffolds, cartilage, etc. [29,30,31,32,33,34]. As aqueous solutions of polymeric materials are generally viscous, a suitable biomaterial ink viscosity is beneficial for controlling the filament size, so most of the studies have characterized the trend of the viscosity of the biomaterial inks with the concentration of compositions or temperatures in detail. However, the results are commonly discrete, and the focus is mainly placed on the printing of biomaterial inks with optimal performance. Therefore, a more systematic investigation of the relationship between the viscosity of gelatin–sodium alginate–methylcellulose biomaterial inks and the composition ratio of materials is still needed, which will be helpful to enable the prediction and modulation of the viscosity of biomaterial inks, and evaluate the printability of biomaterial inks with different viscosities.

On the other hand, a systematic method of experimental design is also needed to predict the rheological properties of biomaterial inks. Most studies adopt a single-factor experimental design to characterize the effect of a single composition, tending to ignore the interaction between different compositions, which may be unable to obtain a globally optimal solution and limit the scope of application of the study. For example, the variety in the hydrogen-bonding network during the co-blending process may lead to unexpected changes in viscosity [35]. The response surface methodology (RSM) can accurately obtain the relationship between each single factor and the response value with the minimal number of experiments through reasonable planning of the experiment sites [36,37]. Andriotis et al. used RSM to explore the relationship between the composition ratio of bioink and extrusion pressure, shrinkage, and water absorption properties to enhance the antimicrobial and wound-healing abilities [38]. Sun et al. optimized the material composition and preparation conditions through RSM, significantly improving the mechanical properties of the hydrogel [39]. Zhou et al. used RSM to build a mathematical model for the degradation properties of hydrogels and then investigated the physiological behavior of cells by preparing hydrogels with different degradation properties [40]. These studies demonstrate the feasibility of RSM in the investigation of the relationship between material composition and certain physical/chemical properties. However, there is still a gap in the study of the relationship between material composition and viscosity. It is necessary to analyze the trend of the rheological properties of biomaterial inks through RSM to provide a basis for biomaterial ink printability studies.

In addition, an important factor that hinders biomaterial inks with different viscosities from printing applications is the lack of optimization solutions for the additive manufacturing bioprinting process, which takes the biomaterial inks as raw materials. Currently, bioprinting tends to use generic additive manufacturing software for pattern design and path planning [41]. However, compared to traditional fused deposition molding (FDM), extrusion-based bioprinting is quite different, particularly in the form of the printed material. The material used in FDM is wire, which is fed through gears, allowing precise control of the feed volume. Moreover, the extruded wire can be pumped back to the print head by the “pull-back” function, which improves the dimensional accuracy and print quality, especially at the starting and ending points of printing [42]. However, the biomaterial inks used in extrusion-based bioprinting are printed on the basis of air pressure, so it is difficult to control the extrusion of biomaterial ink when switching between the printing and non-printing states, which may have an impact on the overall accuracy of the printed structure. The bioprinting process has been investigated widely [14]; however, the research is mostly focused on the optimization of the biomaterial ink composition and concentration, with the optimization schemes and results only applicable to the selected material group, rather than universally practical for other studies in the field of biomaterial inks with specific composition ratios limited by certain properties. Therefore, a general optimization scheme for the improvement of printing accuracy still needs to be explored in depth, which can be applied to biomaterial inks of arbitrary viscosity by simple printing process parameter adjustments.

In this study, gelatin, sodium alginate, and methylcellulose were used as the raw materials of biomaterial ink. The rheological properties were adjusted by controlling the concentration of each composition, and the relationship between viscosity and composition ratios was characterized using RSM. A neural network prediction model was established to further analyze the relationship between the viscosity of biomaterial ink and the quality of printing productions. Printing process optimization was carried out from the perspective of printing pattern path, orienting to biomaterial inks with different viscosities. Combined with RSM, the viscosity properties of biomaterial ink composed of the three constituents mentioned above can be well controlled and predicted in terms of bioprinting with different biomaterial inks. The proposed optimization scheme of the printing process is expected to provide a basis for the design of bio-scaffolds and a reference for systematic printing process development.

## 2. Materials and Methods

### 2.1. Preparation of Biomaterial Ink

Gelatin from porcine skin (type A), sodium alginate (SA, viscosity = 15~25 cp), and methylcellulose (MA, viscosity = 1200~1800 cp) were purchased from Sigma-Aldrich Corp. (Shanghai, China). PBS and CaCl_2_ (AR) were purchased from Biosharp Corp. (Beijing, China) and Sinopharm Chemical Reagent Corp. (Shanghai, China), respectively.

The biomaterial ink composed of gelatin, sodium alginate, and methylcellulose was prepared using a co-blending method. The methylcellulose powder was added to the PBS solution and stirred in a 45 °C water bath. Next, the methylcellulose–PBS solution was placed in an ice-water bath and stirred for 30 min until the methylcellulose was completely dissolved. Gelatin and sodium alginate powders were then added to the methylcellulose–PBS solution and stirred in a 45 °C water bath for 1 h. After that, the biomaterial ink was centrifuged at the rotational speed of 4500 rpm for 5 min to remove air bubbles. The excess biomaterial ink was stored in a refrigerator at T = 4 °C. For simplicity of expression, the involved biomaterial ink in the following section is abbreviated to the corresponding ratios of different components. For instance, the biomaterial ink with a wt% gelatin, b wt% sodium alginate, and c wt% methylcellulose is referred to as a:b:c.

### 2.2. Fourier-Transform Infrared (FTIR)

The successfully prepared biomaterial inks were evaluated using a Nicolet iS50 instrument (Thermal Scientific, Waltham, MA, USA). The biomaterial ink was lyophilized and made into a fine powder. Gelatin, sodium alginate, methylcellulose monomer, and biomaterial ink powder were analyzed separately using the FTIR spectrometer, with a scanning range from 7800 to 350 cm^−1^ and a resolution of 0.09 cm^−1^.

### 2.3. Design of Experiments

To investigate the relationship between the viscosity property of biomaterial inks and the concentration of compositions, the response surface methodology (RSM) was used with the Box–Behnken method and the second-order model chosen for the experimental design. The quadratic polynomial prediction equation was as follows:(1)$$Y=\beta_{0}+\sum \beta_{i}x_{i}+\sum \beta_{ii}x_{i}^{2}+\sum \beta_{ij}x_{i}x_{j}+\varepsilon$$
where *Y* is the predicted value, *β* is the regression coefficient for each item, *x* is the independent variable, and *ε* is the random error.

The concentrations of gelatin (*X*_1_), sodium alginate (*X*_2_), and methylcellulose (*X*_3_) were selected as the independent variables with three levels. The power law was used to model the shear-thinning property of the biomaterial inks by means of the following equation:(2)$$\eta =K{\dot{\gamma }}^{n-1}$$
where *η* is the apparent viscosity, *K* is the consistency coefficient, $\dot{\gamma }$ is the shear rate, and *n* is the power-law index.

The power-law index (n) and the consistency coefficient (K) of the viscosity curve of each biomaterial ink were used as the response values of RSM, expressed as *Y*_1_ and *Y*_2_, respectively. A total of 17 experiments, including 5 center-point replicates, were conducted to ensure the reliability of the model. The available values for independent variables are shown in Table 1.

### 2.4. Rheological Measurements

The viscosity of biomaterial inks at different concentration ratios designed by the RSM was measured using an MCR 72 instrument (Anton Paar, Austria). The solution with a volume of 3 mL was loaded into the gap between the parallel plates of the rheometer, and the excess biomaterial ink was removed with a spatula, leaving the sample equilibrated at T = 30 °C. The shear rate ranged from 1 to 100 s^−1^, and 21 measuring points were selected uniformly in logarithmic coordinates. For the validation experiments of RSM, each test was repeated three times (*n* = 3).

### 2.5. Mechanical Testing

Compression measurements were performed on biomaterial ink samples with a dimension of Φ15 × 5 mm using a WDW-02 electronic universal testing machine (Shidaishijin, China, Shandong, Jinan). Biomaterial ink samples with different concentrations of sodium alginate (e.g., 8:3:3, 8:4:3, 8:5:3) were molded and immersed in 5 wt% CaCl_2_ solution for 30 min to finish the crosslink. The samples with the same concentration ratios but lacking in crosslink were taken as the control group 1. The samples with the different concentrations of gelatin and methylcellulose (e.g., 7:4:3, 9:4:3, 8:4:2, 8:4:4) but crosslinked were taken as the control group 2. During the experiments, the compression rate was fixed at 2 mm·min^−1^, and the compression modulus of each sample was calculated from the relation curve of stress with respect to strain corresponding to the strain range of 0~25%. Each test was repeated three times (*n* = 3).

### 2.6. Latin Hypercube Sampling

Latin hypercube sampling (LHS) was used to generate a spectrum of initial samples for the computational fluid dynamics (CFD) of rheological properties, extrusion pressure, and printing effects. The LHS achieves uniform filling of the entire sampling space by partitioning the input domain and drawing samples randomly in each interval. In this study, the three parameters, including power-law index (0.45~0.7), consistency coefficient (10~100 Pa·s), and extrusion pressure (50~350), were selected to form the vector space through LHS [43]. The samples represented combinations of the biomaterial ink with certain rheological properties and a certain extrusion air pressure.

### 2.7. Modeling and Simulations

The simulations of the printing process were conducted using finite element analysis software (ANSYS Fluent). To comprehensively evaluate the influence of the nozzle on the extrusion process of the biomaterial ink, the model consisting of the front end of the barrel and the tip was constructed on the basis of the measured data, as shown in Appendix A. Since the flow state of the biomaterial ink in the simulation domain is a steady laminar flow [44], the 3D geometric model was simplified to an axisymmetric rotating 2D geometric model. Due to the large size difference between the tip and barrel, in order to improve the accuracy of the simulation, the tip part was encrypted with a size of 0.02 mm during meshing.

Finally, the steady-state simulation of the extrusion of the biomaterial ink at the nozzle was performed using CFD. To reduce the complexity of the CFD simulations, it is usually assumed that the biomaterial ink cannot be compressed, and no slip occurs between the biomaterial ink and the walls of flow channel. Samples obtained from LHS were substituted in the material velocity properties and extrusion pressure parameters. A total of 300 simulation experiments were carried out. The flow velocity distribution of the biomaterial ink at the nozzle was analyzed, and the volume of biomaterial ink extruded per unit time, i.e., the printing flow rate, was obtained through the integration of the velocity curve along the central axis.

### 2.8. Neural Network Prediction Model

In order to reduce the cost of data processing for large-scale samples, the neural network, which can build prediction models for high-dimensional spaces, was chosen as the metamodel, with the prediction performance improved but the size of the model remaining stable as the number of samples increased [45]. The neural network in this study was designed using the back propagation algorithm, choosing the viscosity property of the biomaterial ink and the printing pressure as the input parameters, and the flow rate obtained from the simulations as the output parameter. The data obtained from CFD through the LHS were trained using the algorithm of Bayesian regularization, and the testing data were obtained from the CFD of 50 random points within the specified range.

### 2.9. 3D Bioprinting

The Bio X printer (Cellink, Gothenburg, Sweden) was used to fabricate the designed structures. To ensure the similarity between experimental and simulation conditions, a syringe with a volume of 3 mL and a 25 G needle tip with an inner diameter Φ0.25 mm were used with a fixed layer thickness of 0.2 mm. The printing was performed after stabilization at 30 °C for 20 min to ensure the formation of stable filaments, and the temperature of the printing platform was set at T = 7 °C to induce gelation of the biomaterial ink. Finally, the printed structures were crosslinked by immersing them in 5 wt% CaCl_2_ solution for 5 min.

Based on the print path planning, the bioprinting process was investigated using 8:4:3 biomaterial ink as an example. One-dimensional lines and three-dimensional solid and grid-filled quadrilateral prism structures were selected as the typical prints for process validation. A VHX-1000 digital microscope (KEYENCE, Osaka, Japan) and a CKX53 Culture Microscope (OLYMPUS, Tokyo Metropolis, Japan) were used to assess the fidelity of the prints.

Since 7:4:3 and 9:4:3 biomaterial inks represent low- and high-viscosity biomaterial inks, respectively, the schemes of bioprinting optimization were discussed on the basis of these two composition ratios of biomaterial ink. The grid-filled quadrilateral prism, solid pentagonal prism, hollow cone, and concentric hexagonal structures were printed separately to evaluate the printability of these biomaterial inks. In addition, customized structures printed with both materials together were tested.

### 2.10. Characterization of Microstructures and Printing Processes

The morphological structure of the prints was observed using a scanning electron microscope SU8010 (SEM, HITACHI, Tokyo Metropolis, Japan) at an accelerating voltage of 10 kV. The prints were freeze-dried for 24 h, and sputter-coated with gold before observation. In addition, a charge-coupled device (CCD) camera was used to capture the images of the printing process, especially the extrusion process of biomaterial ink from the nozzle.

## 3. Results and Discussion

### 3.1. Preparation and Characterization of Biomaterial Ink

The printability of biomaterial inks and bioprinting processes have been studied by other researchers [46]; however, different types of materials were commonly used to produce biomaterial inks with different viscosities. In addition, the printing process has been limited to biomaterial inks with a specific range of viscosities [11,46]. It is worth noting that the shear stress increases as the viscosity increases, which may cause cell damage [46,47]. Therefore, it remains necessary to systematically investigate how to control the viscosity of biomaterial inks and the printability of low-viscosity biomaterial inks in order to improve the applicability of the bioprinting process. In this work, the printing processes of high- and low-viscosity biomaterial inks were investigated, respectively, and printing schemes were suggested.

The chemical structures of gelatin, sodium alginate, methylcellulose, and biomaterial ink were characterized using FTIR, as shown in Figure 1A. A C=O stretching vibration of amide I at 1636 cm^−1^ can be observed for gelatin, whereas the asymmetrical and symmetrical stretching vibration of -COO^-^ appears at 1599 and 1409 cm^−1^ for sodium alginate [48,49]. For methylcellulose, the product of partial or complete methylation of the three hydroxyl groups in the glucose of cellulose, the -CH stretching vibration at 2900 and the C-H formation vibration at 1379 cm^−1^ can be observed [50]. In the FTIR spectrum of biomaterial ink, the main bands (2900, 1636, 1599, 1409 cm^−1^) with gelatin, sodium alginate, and methylcellulose can be found, indicating the successful preparation of the gelatin-sodium alginate–methylcellulose biomaterial ink. On the other hand, a change in the size and position of some bands also can be found, e.g., a decrease in the magnitude of the -CH band of methylcellulose. The results suggest that different groups of the materials may interact with each other, which may further increase the hydrogen bonding in the solution [51] and thereby the viscosity of the biomaterial ink [35,52,53].

The rheological behavior of biomaterial inks plays an important role in the extrusion printing process. Figure 1B shows the relationship between viscosity and shear rate for the first five groups of biomaterial inks with different ratios in the RSM. It can be observed that the biomaterial inks showed a significant shear-thinning behavior, with the intercept and slope of the curves varied by changing the concentration of compositions. The shear-thinning behavior of biomaterial inks can help to improve cellular activity during bioprinting, since less damage will be caused to cells when the shear stress on the biomaterial ink reduces. Meanwhile, the fidelity of the printed structure will be enhanced as a result of increased viscosity of the biomaterial ink after it has been left to stand [54]. The force on the biomaterial ink disappears after extrusion, and the viscosity increases, preventing the deformation of the filament, whereas the gelatin is crosslinked at low temperatures, which keeps the structure in shape. Therefore, a careful selection of biomaterial ink with the proper rheological properties has an important impact on biocompatibility and structural fidelity. However, most research on extrusion-based printing mainly focuses on the development of specific materials for biocompatibility and physiological function induction [55,56,57], ignoring the control of viscosity properties from a physicochemical perspective, which reduces the ability to improve bioactivity through the printing process.

### 3.2. Analysis of the RSM Model

To fully understand the effect of composition concentration on the viscosity property of biomaterial ink, the RSM was used to establish a mathematical model. Statistical data and relationships were obtained between the power-law index, consistency coefficients, and the gelatin–sodium alginate–methylcellulose biomaterial inks with different concentration ratios, respectively. The experiment’s design and results are shown in Appendix A, and the ANOVAs for the power-law index and consistency coefficient are shown in Appendix A. To simplify the expression, gelatin, sodium alginate, and methylcellulose are abbreviated to Gel, SA, and MC, respectively, in this section.

The variance was analyzed to verify the validity of the RSM model. For the power-law index (Appendix A), the coefficient of determination (R-squared, R^2^), adjusted R^2^, and predicted R^2^ were 96.73%, 94.18%, and 91.02%, respectively, indicating a good fit of the model to the data. That is, the present value of R^2^ indicates that the model could explain 96.73% of the variability in the response, while the value of the predicted R^2^ implies a high predictive power of the model. The *p*-value for the model was less than 0.05, indicating that the model term is significant. In this case, the Gel, SA, MC, Gel × Gel, and Gel × MC terms were significant (*p* < 0.05), indicating that they play an important role in controlling the power-law index. In contrast, the SA × SA and MC × MC terms were not significant (*p* > 0.05), but their retention was beneficial for improving the R^2^, i.e., the prediction performance of the mathematical model when the RSM was optimized by the backward method. The *p*-value of 0.86 for the misfit term of the model indicates that this term is not significant and that the model is adequate. Similarly, for the consistency coefficient, the R^2^, adjusted R^2^, and predicted R^2^ were 99.38%, 98.58%, and 91.78%, respectively. The model term was significant (*p* < 0.05), while the misfit term was not significant (*p* > 0.05), indicating the usability and adequacy of the model. Since the model was proven to possess good validity and feasibility, it could be used for the subsequent analysis and prediction of the power-law index and consistency coefficient of biomaterial inks. The results demonstrate that the viscosity properties of the biomaterial ink are significantly related to the concentration ratios of the composition in this material group.

The statistical results were analyzed in order to fully understand the effect of the concentration ratios of the biomaterial ink on the power-law index and the consistency coefficient. From the point of data statistics, Pareto diagrams of the power-law index and consistency coefficient with the biomaterial ink compositions were created first. As shown in Figure 1C, the standardized effect of the composition concentration ranked in the order of Gel > MC > SA > Gel × Gel > Gel × MC > SA × SA > MC × MC, indicating a dominant position of gelatin. Moreover, the results for the primary terms were all higher than those of the secondary terms, suggesting that the effect of material concentration on the power-law index is more significant than the interaction between materials; however, the role of the secondary terms is still not negligible. The most significant effect of the concentration of gelatin on the power-law index may be ascribed to the correlation found between temperature and the power-law index [58], while polymeric materials usually show temperature sensitivity in aqueous solutions [59]. Furthermore, compared to the other compositions, gelatin is the most sensitive to temperature, inducing entanglement and crosslinking of molecular chains as the temperature decreases. Therefore, it is reasonable that the concentration of gelatin has the most significant effect on the power-law index. As shown in Figure 1D, the distribution of statistical results for the consistency coefficient is similar to that of the power-law index. That is, the effect of the concentration of gelatin was the most significant, and all primary terms were higher than the secondary terms, indicating that the concentration of the polymer materials has a high effect on the consistency coefficient [60]. However, compared to the power-law index, Gel × SA, Gel × MC, and Gel × Gel showed a relatively high effect on the consistency coefficient, which may be explained by the interactions between intermolecular functional groups of polymeric materials. That is, as shown in FTIR spectroscopy (Figure 1A), the mixture of three materials leads to changes in hydrogen bonding. In summary, it is still necessary to consider the effects of material properties and interactions between different compositions in a comprehensive manner in the investigation of the rheological properties of biomaterial ink.

To predict the viscosity property of the biomaterial ink, a mathematical model was developed on the basis of the statistical results, as shown in Equations (3) and (4).
n = −0.256 + 0.1638 × Gel + 0.1596 × SA + 0.1828 × MC − 0.01107 × Gel × Gel − 0.0259 × SA × SA − 0.0213 × MC × MC − 0.01348 × Gel × MC(3)
K = 2014481 − 277312 × Gel − 345128 × SA − 329307 × MC + 10161 × Gel × Gel + 18857 × SA × SA + 19065 × MC × MC + 22029 × Gel × SA + 21509 × Gel × MC + 22724 × SA × MC(4)

To visualize the above model, the different ratios of the biomaterial ink compositions within the preset range were enumerated, with the results plotted as a two-dimensional image in scatter form (Figure 1E). It can be seen that the viscosity property of the biomaterial inks covered a region approximating a trapezoid, with an overall distribution trend from a low power-law index and high consistency coefficient to a high power-law index and low consistency coefficient. In addition, when the power-law index of the biomaterial ink exceeded a certain threshold, the change in consistency coefficient tended to level off, i.e., the small region outside the fitting trapezoid. Therefore, the current material group could achieve a wide selection range of viscosity properties for biomaterial inks, which is conducive to the subsequent investigation of printing process experiments with respect to the viscosity properties of biomaterial inks. In addition, the present material group may also be used for pre-experimental exploration of the printing process on expensive materials. For instance, the material group can be used to simulate the rheological properties of expensive biomaterial inks, which will help to optimize the process parameters for printing applications of expensive biomaterial inks without the high costs.

Finally, a series of experiments were conducted to measure the viscosity properties of the biomaterial inks, in order to verify the validity of the model. In addition to the existing ratios of the RSM, the viscosity properties of five randomly selected biomaterial inks were tested, and the results are shown in Table 2. It can be found that the prediction accuracy of the power-law index (n_Accuracy_) was higher than 96%, while the prediction accuracy of the consistency coefficient (K_Accuracy_) was higher than 79% for all the available biomaterial inks, which clearly confirms that the proposed mathematical model shows a reliable prediction ability.

### 3.3. Analysis of the Mechanical Property

The mechanical property is one of the most important physical properties of biomaterial inks, which will impact the behavior of seeded cells [61,62]; therefore, the evaluation of the effect of composition concentration on the mechanical properties of biomaterial ink is quite necessary. Currently, the bioprinted scaffold has shown potentially promising applications in in vivo experiments, in order to replace damaged cartilage or tissues [63,64], which needs the prints to possess similar mechanical properties with the surrounding tissues and to be able to withstand the compressive stresses. In addition, some studies have shown that the compressive stress in the microenvironment may induce cell physiological behaviors (e.g., cell differentiation, adhesion, migration, etc.) [65,66], which will be the reference for the studies on scaffolds with gradient mechanical properties. Therefore, the compression modulus is believed to be a critical parameter in characterizing the mechanical properties of biomaterial inks. Since the ionic crosslinking occurring between sodium alginate and calcium ions has a significant effect on the mechanical properties of biomaterial inks [67], the compression modulus of biomaterial inks before and after crosslinking was tested under different conditions (The concentrations of sodium alginate are set to be 8:3:3, 8:4:3, 8:5:3, respectively), as shown in Figure 1F. It can be observed that the compression modulus of the crosslinked biomaterial inks (0.39 ± 0.077, 0.54 ± 0.088, 0.66 ± 0.111 kPa) was significantly enhanced, as compared to the biomaterial inks without crosslinking (0.13 ± 0.025, 0.13 ± 0.006, 0.16 ± 0.006 kPa). The compression modulus of the crosslinked biomaterial inks increased with increasing concentrations of sodium alginate, whereas those of the non-crosslinked biomaterial inks showed no significant change. The results indicated that the mechanical properties of the biomaterial inks could be effectively controlled by crosslinking the biomaterial inks with different concentrations of sodium alginate. Further, the effect of concentrations of gelatin and methylcellulose on the compression modulus of biomaterial inks was explored. The values of the compression modulus of biomaterial inks with different concentrations of gelatin or methylcellulose are shown in Appendix A. It can be found that the compression modulus (0.47 ± 0.043, 0.54 ± 0.088, 0.64 ± 0.018 kPa) of the biomaterial inks increased with the rising concentration of gelatin (7:4:3, 8:4:3, 9:4:3). This is because the increase in gelatin concentration may enhance the interpenetrating network with sodium alginate, finally resulting in a higher compression modulus of the biomaterial inks. In contrast, the changes in the concentration of methylcellulose (8:4:2, 8:4:3, 8:4:4) did not show any direct correlation with the compression modulus (0.52 ± 0.043, 0.54 ± 0.088, 0.53 ± 0.053 kPa), indicating that the rheological properties of biomaterial inks in this material group can be adjusted by controlling the concentration of methylcellulose to avoid loss of the mechanical properties. That is, the printability of the biomaterial ink with the desired mechanical properties, which are determined by gelatin or sodium alginate, can be improved by adjusting the concentration of methylcellulose. In previous studies, single- or dual-material composition biomaterial inks were adopted to investigate the effect of mechanical properties on the physiological behavior of cells [68,69]. While regulating the concentration of sodium alginate to control the mechanical properties of the biomaterial ink, it inevitably changes the rheological properties of the biomaterial ink, which may reduce the printability and lead to the inability to perform printing experiments. This work may provide a new research idea to achieve controlled regulation of single properties of a biomaterial ink by synthetically adjusting the concentration of compositions, according to the impact effect of each composition on the properties of the biomaterial ink.

### 3.4. Analysis of Simulation Results

CFD was used to investigate the extrusion flow of biomaterial inks with different viscosity properties under varied extrusion pressures. The results showed that the mean value of skewness was 0.16, indicating that the shape of most of the mesh cells is close to an equiangular quad, which is conducive to improving the finite element simulation performance of the model. Simulation tests were performed on the 8:4:3 biomaterial ink to exhibit the extrusion state at pressures of 200, 250, and 300 kPa. Figure 2A shows the pressure clouds along the axial cross-section for the entire simulation domain. It can be seen that the pressure varied drastically at the needle tip but seemed constant in the barrel section, due to its larger cross-sectional area than the syringe section. Figure 2B gives the velocity distribution at the nozzle, showing that the flow rate increased as the extrusion pressure was raised. Meanwhile, the flow rate of the biomaterial ink close to the wall was obviously lower than that located in the neighborhood of the central axis. This is due to the fact that the biomaterial ink is a power-law fluid with friction existing between the layers of the fluid, as shown in Figure 2A [70].

### 3.5. Analysis of the Prediction Model

During application-oriented bioprinting research, the biomaterial ink composition and printing parameters need to be continuously adjusted to meet the performance and metrics of the print model, which requires a lot of time and effort. Therefore, this study analyzed the change in flow states through systematic simulation. In order to observe the viscosity properties of biomaterial ink, which are difficult to regulate in experiments, and the extrusion pressure parameter, which is frequently regulated, the power-law index, consistency coefficient, and extrusion pressure were selected as input parameters. Since the parameters for the extreme cases cannot be applied in actual bioprinting, it is necessary to ensure the simulation parameters match well with the ones in actual printing, in order to guide the design of the printing process on the basis of the simulation results. Therefore, the range of the power-law index and consistency coefficient for the simulations needs to be determined first to avoid resource wasting. Considering the printability of the biomaterial ink [71] and the number of the simulation samples, a range of 10~100 Pa·s was chosen for the consistency coefficient. Combined with the RSM results, the power-law index of the biomaterial inks was suggested to be distributed in the range of 0.45~0.7 for the determined consistency factor.

In computer simulation experiments, generating random errors is not usual, and there is no need to repeat the experiment. However, the generation of systematic errors still seems inevitable; thus, more emphasis needs to be placed on experimental points that can fill the entire design space [72]. In this work, Latin hypercube sampling (LHS) was adopted, and the flow states of all samples were simulated by CFD. In order to improve the prediction efficiency and reduce expenditures, a PB neural network was used to analyze and predict the results of CFD.

For the selection of the output variables, it is assumed that the extruded biomaterial ink adhered to the printed scaffold at the starting point and moved along with the printer head at the terminal point during the printing process, thus being stretched uniformly into a desired cylinder. The diameter of the filament, which can be obtained by calculating the flow rate and printing speed (Equation (5)), was usually adopted to characterize the print quality.
(5)$$d=2\sqrt{\frac{Q}{\pi v}}$$
where *d* is the diameter of filament, *Q* is the flow rate, and *v* is the printing speed.

The flow rate can be obtained by integral calculation of the velocity distribution curve at the nozzle obtained from the simulation results. Once the flow rate is determined, it is possible to quickly select the appropriate printing speed through calculations and to predict the diameter of the filament reliably. Therefore, the flow rate was selected as the output variable for the PB neural network in this work.

To validate the PB neural network model, additional 50 sets of random parameters were selected for the CFD, and the prediction results from the neural network model were compared with the CFD results. As shown in Figure 2C, the biomaterial ink flow rate obtained from the CFD simulation was randomly distributed in the interval of 0~4 μL, and the neural network’s predicted values accurately reproduced the corresponding results. The error values shown as the blue dots were distributed between ±0.05 with no obvious pattern. Further, two common parameters, i.e., mean absolute error (MAE) and root mean square error (RMSE), were used to evaluate the prediction accuracy of the model.
(6)$$MAE=\frac{1}{n}\sum_{i=1}^{n}\left|y_{i}-{\hat{y}}_{i}\right|$$
(7)$$RMSE=\sqrt{\frac{1}{n}\sum_{i=1}^{n}{\left(y_{i}-{\hat{y}}_{i}\right)}^{2}}$$
where *n* is the number of samples, *y* is the observed value, and $\hat{y}$ is the predicted value.

MAE and RMSE indicate the mean of the absolute error and the standard deviation of the difference between the predicted and observed values, respectively [73]. In the work, the MAE was 0.60%, and the RMSE was 0.81% (Appendix A in the Supplementary Information), both of which were less than 1%, indicating that the prediction values of the neural network model agree well with the CFD results; therefore, the neural network prediction model could be a good substitute for CFD to predict the flow rate.

Figure 2D shows the results of the neural network prediction model of the power-law index and consistency coefficient of the biomaterial ink on the flow rate under a particular extrusion pressure. It can be seen that the increase in the power-law index reduced the flow rate. This phenomenon probably can be ascribed to the fact that the viscosity curve of the biomaterial ink tends to flatten as the power-law index increases (Figure 2E), exhibiting higher viscosity and thus reducing the flow rate. An increase in the consistency coefficient also reduced the flow rate. It can be noticed that when the power-law index and the consistency coefficient of the biomaterial inks are both kept low, the surface was relatively steep, implying that a small change in the viscosity properties of the biomaterial ink may lead to large fluctuations in the flow rate at the same print pressure. This is because the biomaterial ink within the above characteristic region may be too thin to be successfully printed, resulting in a tendency to collapse. Therefore, the studies of the printing process of this specific biomaterial ink may not be reproducible and generalizable either. Conversely, the rest of the region showed low flow rates of the biomaterial ink with slight variations, and the tolerances generated in the preparation of the biomaterial ink may have relatively little impact on the printing process.

To further characterize the relationship between the viscosity property of the biomaterial ink and the flow rate, the k-means clustering, which is an unsupervised pattern recognition method to classify data based on the similarity of certain features [74], was used to analyze the surfaces formed by the flow rate with the power-law exponent and consistency coefficient. In this work, the surface was divided into three groups, representing high-viscosity, medium-viscosity, and low-viscosity biomaterial inks, respectively, which are indicated by yellow curves in Figure 2D. It can be found that the overall distribution of the three separate groups was continuous. In the high-viscosity biomaterial ink group, there was less variability in the points, suggesting that the printing process for biomaterial inks within this region may be more generic. In the medium-viscosity group, the biomaterial inks showed relatively moderate rheological properties, but the high variability in flow rate may restrict the design of a universal printing process. In the low-viscosity group, the biomaterial inks showed high flow rates, which were less printable or required special printing schemes and process designs. In addition, as shown in the green circle in Figure 2D, the boundaries of groups remained slightly variable under different pressures (P = 150, 200, 250, 300 kPa), which may be influenced by the apparent viscosity of the biomaterial ink during the extrusion process.

Figure 2E shows the evolution of the flow rate with varying extrusion pressure for different power-law indexes of the biomaterial ink with the same consistency coefficient (K = 55 Pa·s), whereas Figure 2F shows the evolution of the flow rate with varying extrusion pressure for different consistency coefficients of the biomaterial ink with the same power-law index (n = 0.575). It can be noticed that the flow rate increased with the increasing extrusion pressure in a linear trend. The extrusion pressure–flow rate curves of biomaterial inks with different power-law indices were more consistent than those with different consistency coefficients. Therefore, it may be possible to improve the versatility of the printing process by regulating the consistency coefficients of biomaterial inks into the roughly same value.

### 3.6. Analysis of the Printing Process

The purpose of this work is to explore the relatively universal bioprinting process, providing references and suggestions for future research to make the best use of commercially available biomaterials. As is well known, the printing parameters and printing process determine the shape of the filament, which will further affect the dimensional accuracy of the prints. Therefore, the effect of extrusion pressure and printing speed on the width of the filament was investigated separately using a single-factor experimental method. The biomaterial ink with a concentration ratio of 8:4:3 was used in the experiments. As shown in Figure 3A, the width of filament increased with ascending extrusion pressure in an approximately linear trend. By contrast, the width of the filament decreased with increasing printing speed in an approximately non-linear trend. It is clear that extrusion pressure and printing speed can work together to control the width of the filament. However, when the printing speed was low (≤12 mm/s), the width of the filament declined rapidly as the printing speed increased, but the evolution curve tended to flatten out when the printing speed was higher than 12 mm/s. In contrast, the effect of printing pressure on the width of the filament was more stable. In addition, a high printing speed is helpful to increase the printing efficiency; however, too high a printing speed may cause instability in the structure of the print due to the limitations of the printer hardware. In this work, a printing speed of 12 mm/s and an extrusion pressure ranging from 200 kPa to 300 kPa were adopted.

To investigate the effect of printing pattern path planning on the dimensional accuracy of the filament, the single line was designed with common path planning, and the lengths of the printed filaments were measured (Table 3). It can be seen that the measured lengths of the filaments (unoptimized) are significantly smaller than the design values, which may be ascribed to the fact that the yield stress hinders the flow of biomaterial ink at the beginning of printing [75]. In multi-layer printing, this error may accumulate, leading to defects in the prints. Therefore, a variable-printing speed method was developed for process improvement. When the nozzle switched from a non-printing state to a printing state, the printing speed would be changed to 1 mm/s, lasting for a certain time (*t*_0_) before returning to the normal printing speed (12 mm/s), as shown in Figure 3B. The widths of the filaments printed at 1 mm/s for 0.1 s, 0.2 s, 0.3 s, 0.4 s, and 0.5 s were tested separately (Figure 3C), and the length of filament began to show a small error when *t*_0_ = 0.1 s. As the duration increased, the build-up of biomaterial ink at the starting point was increasingly aggravated, which in turn reduced the print quality. A CCD camera was therefore used to monitor the bioprinting process (Figure 3D), observing that the biomaterial ink flowed out of the nozzle and expanded outward at the beginning of printing, and then flowed downward to form a stable filament after a period of time. This may be explained by the phenomenon of extrusion swelling that occurs during the extrusion process, resulting in significantly larger diameters or thicknesses of the extrudate than the diameter of the polymer extruded from a tiny orifice [76]. Although this phenomenon seems negative on the printing accuracy, it has been confirmed by experiments that the errors can be effectively improved by controlling the delay time of 0.1 s for 1 mm/s printing. On the other hand, a small build-up of biomaterial ink at the start can improve the adhesion of the next layer, further improving the accuracy and quality of the print. Therefore, *t*_0_ was set to be 0.1 s, and the normal printing speed was changed from 12 mm/s to 8 mm/s and 16 mm/s, respectively, to verify the applicability of the optimization solution of printing path. It can be clearly seen from Table 3 that the accuracy of the length of the optimized filaments is obviously higher than the unoptimized filaments. That is, the optimization solution is capable of reducing the dimensional error of the printed filaments.

In the path planning of conventional extrusion additive manufacturing, the boundary of the model is usually printed first, followed by the filling of the inner area at a certain density. The overlap between the inner filling and the outer boundary is vital for the printing path planning of 3D structures, and the ratio of the overlap width to the width of a single filament is defined as the overlap ratio. In commercial software, the ratio of the overlap width is usually set to 50% by default; however, there is still a need to further explore the rationality of “overlapping” in the bioprinting process in order to further improve the precision and quality of biomaterial ink prints. Therefore, this study evaluated the effect of different overlap states on the structural quality of prints.

A 0/90° grid-filled quadrilateral prism was designed with varied spacing between neighboring lines within different domains (inner filling lines: 0.8 mm; spacing between the outer frame and the fill pattern: 0.5 mm). As the overlap ratio could not be calculated explicitly in bioprinting, the distance from the short edge of the grid fill to the boundary was used instead in this work, which was *D*_2_ set to 0.5 mm, 0.4 mm, 0.3 mm, 0.2 mm, 0.1 mm, and 0 mm, respectively. The model composed of three layers was printed with the same pattern for the first and second layers, but a different pattern (Figure 3E) for the third layer. Figure 3F shows that when the distance was 0.5 mm or 0.4 mm, the filament formed a semicircular corner, with only part of the biomaterial ink printed in the specified position. That is, the short edge shifted inwards, deviating from the designated position. This phenomenon may have arisen from too high a printing speed and too short an edge, resulting in the fact that the nozzle started to change direction before the extruded biomaterial ink was deposited in the designated position and finished crosslinking. Therefore, the extruded biomaterial ink at this point deformed and gradually merged into a thicker filament because the pulling of the nozzle was greater than the adhesion force from the second layer of filament. Since the filaments of a short edge cannot be deposited in the correct position, the error will be accumulated to worsen the printing of subsequent filaments. For the distances ranging from 0~0.3 mm, the short edge adhered to the outer frame and deposited correctly on the outermost supporting filament. In contrast to the previous prints, the extruded biomaterial ink in these patterns could adhere to the outer frame even though it had not yet been fully deposited on the support lines. The reason may be that the short edge was exerted by the adhesion force from both the support lines and the outer frame, which was able to balance the pulling from the nozzle, finally achieving accurate printing of the filament (outlined by the orange circle in Figure 3F). However, when the short edge was too close to the outer frame (at the green circle in Figure 3F), a significant expansion occurred at the ends of the short edge because the outer frame had been crosslinked and shaped, limiting the deposition of biomaterial ink and thereby producing upward building of the extruded biomaterial ink. According to the experimental results in this work, the optimal distance from the short edge to the outer frame was selected to be 0.2~0.3 mm.

To verify the effect of process optimization, a solid and a grid-filled quadrilateral prism with dimensions of 10 × 10 × 3 mm^3^ were printed separately before and after the optimization of the printing path. As shown in Figure 4A, the optimized prints had a higher degree of structural integrity, whereas obvious defects appeared in the unoptimized solid quadrilateral prism at the start, which had been highlighted by the red circle. The defective structure may have poor mechanical properties and inadequate contact at the interface with other structures. Similarly, the fidelity of the unoptimized grid-filled quadrilateral prism seems quite poor due to the inward shift of the grid filling, which causes the biomaterial ink to accumulate at the nozzle instead of being deposited in the specified position. Solid quadrilateral prisms with different dimensions (5 × 5 × 3 mm^3^ and 15 × 15 × 3 mm^3^) were also fabricated (Figure 4A) to verify the planning capability of the printing process for models of different sizes. It can be observed that the prints had a high dimensional accuracy with good structural integrality and surface evenness. Meanwhile, the cross-sections of the printed filaments were uniformly shaped with an intact structure (Figure 4B). These results demonstrated the effectiveness of the designed printing path optimization scheme in the bioprinting of typical models with sufficient fidelity.

To investigate the effect of crosslinking on the accuracy of the filaments, grid-filled quadrilateral prisms with four layers were printed with 8:3:3, 8:4:3, and 8:5:3 biomaterial inks, respectively. The prints were crosslinked in 5 wt% CaCl_2_ for 5 min; the widths of filaments were measured before and after crosslinking, and the shrinkage ratios were calculated, as shown in Appendix A. The results showed that the filaments did not deform significantly after crosslinking in the grid-filled quadrilateral prism printed with 8:3:3 biomaterial ink (shrinkage ratio = −8.7%), whereas the shrinkage ratio of filaments printed with 8:4:3 and 8:5:3 biomaterial inks were 12.3% and 9.9%, respectively. The chelating reaction between sodium alginate and Ca^2+^ during crosslinking may lead to the shrinking of the network structure, while the swelling of the biomaterial inks may result in the expansion of the filaments. When the concentration of sodium alginate remains low or high, the above two processes cannot be dynamically balanced, resulting in the perceptible expansion or shrinking of the filaments. To reduce the deformation of the prints, more impact factors, for instance, concentration of CaCl_2_, crosslinking duration, crosslinking agent, etc., need to be involved in the investigation of the crosslinking process and the swelling characteristics of biomaterial inks. In addition, to evaluate the effect of crosslinking on the entire structure of the prints, a solid and a grid-filled quadrilateral prism with dimension of 10 × 10 × 3 mm^3^ were printed with 8:4:3 biomaterial ink, respectively (Appendix A). It can be found that the entire structure matched well with the design goals, showing good printing quality at the short edges and starting points. Although the filaments undergo minor expansion and shrinking during crosslinking and swelling, the entire structure printed by optimized printing processes presents high fidelity.

In addition, the grid-filled quadrilateral prism was examined using SEM to analyze the microscopic morphology. As shown in Figure 4B, pore structures distributed uniformly were present in the cross-sections, which is beneficial for cell attachment and nutrient exchange with the culture medium [77].

It is important to pay attention to the limitations of preparing and printing biomaterial inks with different viscosities. When the viscosity of the biomaterial ink is too low, the printing flow rate may be difficult to control to achieve accurate fabrication of the prints. The flowing of the biomaterial ink before the crosslinking and shaping of gelatin may also reduce the fidelity of the prints [46], which needs to be modeled to restrain the penetration and spreading of low-viscosity biomaterial inks. On the contrary, when the viscosity of the biomaterial ink is too high, it seems difficult to be extruded despite the shear thinning [78], leading to frequent blocking within the needle. In addition, the biomaterial inks with high viscosities may require powerful equipment and additional processes for mixing and defoaming during preparation. To evaluate the applicability of the optimized printing path scheme to biomaterial inks with low and high viscosity, respectively, the biomaterial inks with ratios of 7:4:3 and 9:4:3 were tested. The predicted values for the viscosity property of the selected biomaterial inks were, respectively, n_7:4:3_ = 0.65, K_7:4:3_ = 17.24 Pa·s; n_9:4:3_ = 0.54, K_9:4:3_ = 93.05 Pa·s. The 7:4:3 biomaterial ink was used to fabricate a grid-filled quadrilateral prism, a solid quadrilateral prism, a solid pentagonal prism, and a pyramidal structure, respectively. As shown in Figure 5A, the collapse of the grid-filled quadrilateral prism was significant, but the solid structure showed better fidelity, while the pyramid structure exhibited a high degree of fidelity at the start of each layer (as highlighted by the blue circle). A hollow cone structure, a pyramidal structure, a concentric hexagonal structure, and a grid-filled quadrilateral prism were printed, respectively, using 9:4:3 biomaterial inks. As shown in Figure 5B, all the prints had high fidelity with intact structures, confirming that a high-viscosity biomaterial ink could be capable of printing complex structures, such as overhanging and thin-walled structures, by means of the optimized pattern path.

From above, it can be seen that both biomaterial inks showed good fidelity in the fabrication of solid structures, probably arising from the fact that the underlying glass plate or biomaterial ink can provide support for the biomaterial ink being extruded, thereby slowing down the collapse rate. Meanwhile, the material group in this work presents good low-temperature crosslinking and shear-thinning properties, allowing even low-viscosity biomaterial inks to quickly solidify with certain mechanical properties after extrusion. However, in the case of the grid-filled models, the lower viscosity biomaterial inks are unable to form suspended structures, whereas the higher viscosity biomaterial inks are still good at producing prints with better structural quality. In addition, the low-viscosity biomaterial inks could be extruded at lower extrusion pressures, allowing the spreading of extruded biomaterial inks optimizable by the effective control of extrusion pressure. On the contrary, the high-viscosity biomaterial inks required a higher extrusion pressure with less deformation of the biomaterial ink after extrusion. Therefore, the high-viscosity biomaterial ink can be considered the skeleton, and the low-viscosity biomaterial ink can be used for filling or loading cells. As shown on the left in Figure 5C, the solid structure was printed quickly using low-viscosity biomaterial inks, which was capable of providing support for the superstructure after low-temperature crosslinking with greater printing efficiency. On the right in Figure 5C, low-viscosity biomaterial ink was printed on a dense parallel-line fill by designing the path of model appropriately, which could be used for biological functions, such as cell loading. The structure with a “HIT” pattern was printed and displayed in Figure 5D, showing that the low-viscosity biomaterial ink did not spread outwards significantly under the constraints of the grid structure and the low-temperature crosslinking. Compared to the studies using biomaterial inks and artificial polymers (e.g., PCL, PLA, PBS, etc.) for co-printing [79], better biocompatibility may be achieved by co-printing with biomaterial inks of different viscosities.

## 4. Conclusions

In this study, to systematically investigate the relationship between the rheological properties of biomaterial inks and printability, gelatin, sodium alginate, and methylcellulose were used as the material groups of biomaterial ink to reveal the relationship between the material composition ratio and the viscosity, with the model for predicting the viscosity property of the biomaterial ink also developed. The statistical results showed that the proposed model had high significance and good predictive ability. The power-law index and consistency coefficient of biomaterial inks can be predicted with an accuracy of 96% and 79%, respectively, and the number of required experiments is significantly reduced compared to conventional experiments. The prediction model for flow rate at the nozzle was established by CFD and neural network, and the influence of the viscosity property of the biomaterial ink and extrusion pressure on the flow rate was discussed. Based on the flow rate trend, the biomaterial inks were categorized into three viscosity levels: low viscosity, medium viscosity, and high viscosity. The fluctuation of flow rate with the viscosity property seems minor in the region of high viscosity, allowing for a high degree of versatility in the printing process. The switching between different extrusion states and the overlap pattern were optimized by means of printing path planning. The accuracy of the length of the filaments can be improved to 99% by introducing a 0.1 s motion delay at the starting points. A suitable overlap between the inner filling and the outer boundary (spacing of 0.3 mm) was verified to be capable of significantly improving the quality of the prints. Finally, the printing schemes for the biomaterial inks with low and high viscosity were explored, providing a strong reference for the design of the printing process. However, the approach presented in this study mainly focuses on material properties and printing processes.

Although the printability of the biomaterial ink and the printing process play critical roles in the bioprinting process, it is still necessary to evaluate the biocompatibility of the materials in order to approach the level of application in subsequent studies, such as the differences in cellular activity and functional expression through different composition ratios of biomaterial inks. In addition, the relationship between material microstructure, other properties of biomaterial inks (e.g., yield stress, creep, thixotropy, etc.), and the printability of biomaterial inks can be explored in depth to further design printing schemes with a wider range of biomaterials.

## Figures and Tables

**Figure 1 bioengineering-10-01358-f001:**
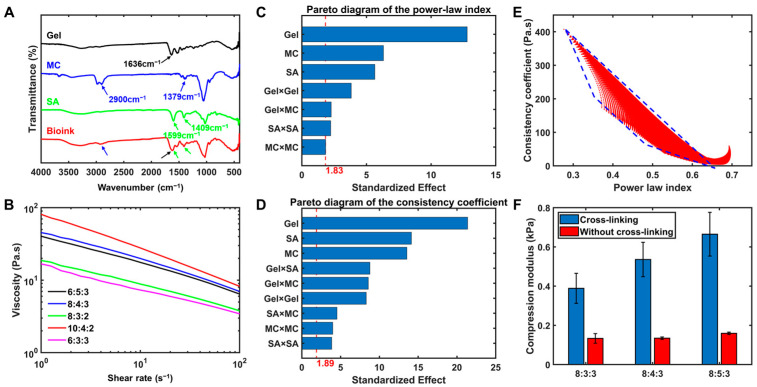
(**A**) FTIR spectra of gelatin, sodium alginate, methylcellulose, and biomaterial ink. (**B**) Relationship of viscosity with shear rate of biomaterial inks in RSM. (**C**) Pareto diagram of the power-law index of RSM. (**D**) Pareto diagram of the consistency coefficient of RSM. (**E**) Distribution of rheological properties for the biomaterial ink with different ratios of compositions. (**F**) Compression modulus of biomaterial inks with three typical composition ratios (8:3:3, 8:4:3, and 8:5:3) before and after crosslinking.

**Figure 2 bioengineering-10-01358-f002:**
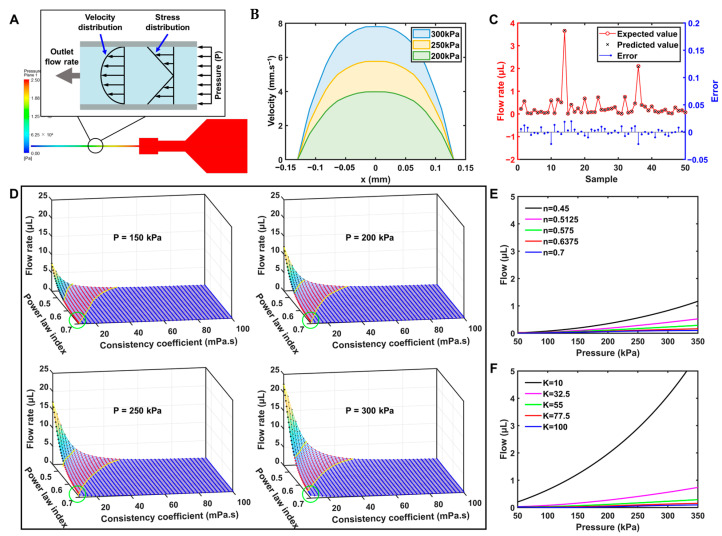
(**A**) Pressure distribution of the biomaterial ink in the simulation domain under the printing pressure of 250 kPa, and mathematical model of the flow for a shear-thinning fluid in a tubular channel. (**B**) Velocity distribution at the nozzle under different extrusion pressures. (**C**) The results and errors of CFD and neural network simulation. (**D**) Relationship of flow rate with rheological properties of biomaterial ink under different printing pressures. (**E**) The evolution of the flow rate with extrusion pressure for biomaterial inks under the condition of different power-law indices, but the same consistency coefficient (K = 55 Pa·s). (**F**) The evolution of the flow rate with extrusion pressure for biomaterial inks under the condition of different consistency coefficients, but the same power-law index (n = 0.575).

**Figure 3 bioengineering-10-01358-f003:**
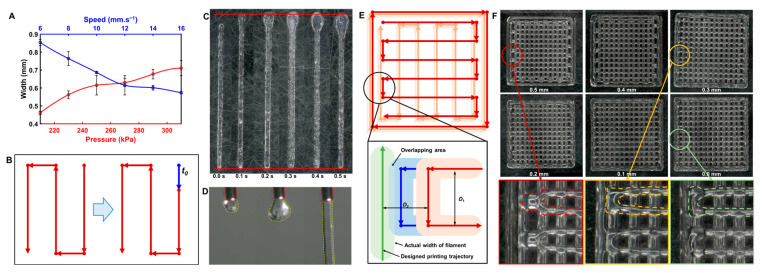
(**A**) Relationship of filament width to extrusion pressure and printing speed. (**B**) The printed results of the filaments at different *t*_0_. (**C**) Optimization scheme for the starting stage of bioprinting. The blue line segment corresponds to a printing duration of t0 at low printing speed (1 mm/s), followed by the residual segment at normal printing speed. (**D**) The extrusion state of the biomaterial ink at the nozzle monitored by a CCD camera. (**E**) Schematic diagram of the overlap between the inner filling and the outer boundary. The colors from light to dark represent the first, second, and third layer traces, respectively. In the below-corner inset, the green color represents the outer boundary, and the red and blue colors represent the two cases of non-overlap and overlap, respectively. (**F**) The printing results with different distances between the inner filling and the outer boundary.

**Figure 4 bioengineering-10-01358-f004:**
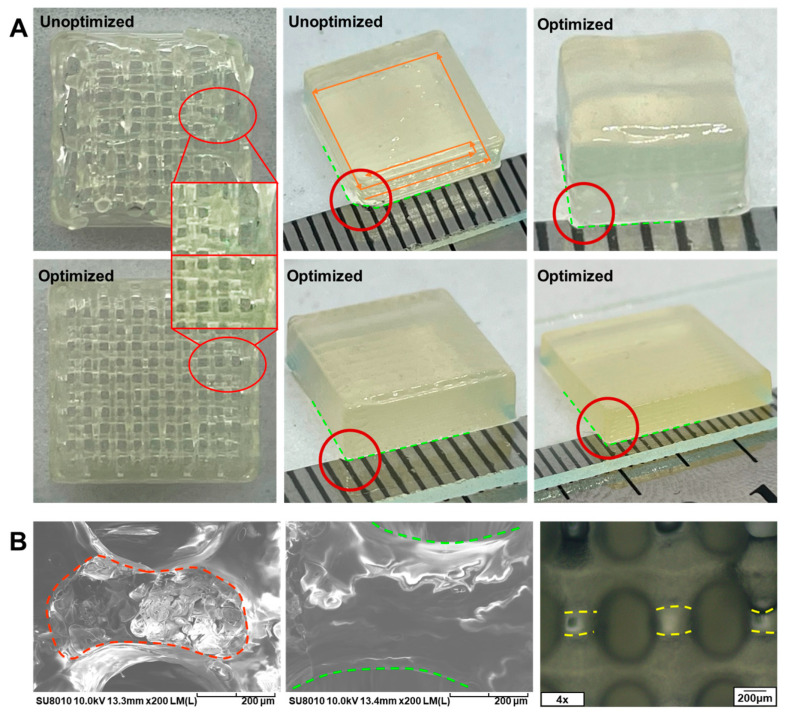
(**A**) Grid-filled and solid quadrilateral prism of different sizes (5 × 5 × 3, 10 × 10 × 3, 15 × 15 × 3 mm) printed before and after path optimization. (**B**) SEM of filament section and surface, as well as scaffold pores observed under an optical microscope (The outlines of filament section, filament, and scaffold pores have been highlighted by the red, green, and yellow lines, respectively).

**Figure 5 bioengineering-10-01358-f005:**
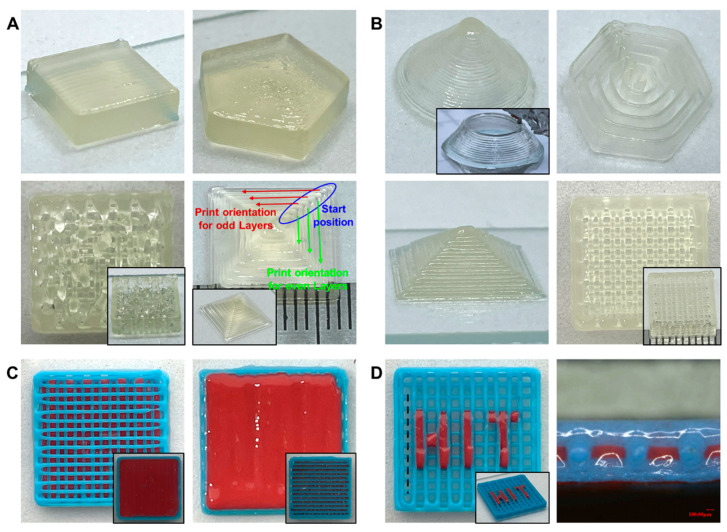
(**A**) Different structures (solid quadrilateral prism, grid-filled quadrilateral prism, solid pentagonal prism, and pyramid) printed using 7:4:3 biomaterial inks. (**B**) Different structures (hollow cone, pyramid, concentric hexagon, and grid-filled quadrilateral prism) printed using 9:4:3 biomaterial ink. (**C**) Grid-solid structures co-printed using 7:4:3 and 9:4:3 biomaterial inks. In the left figure, the bottom supporting layer was printed using 7:4:3 biomaterial ink (red), and the top grid was printed using 9:4:3 biomaterial ink (blue). The inset image is viewed from the bottom up. In the right figure, the bottom grid is printed using 9:4:3 biomaterial ink (blue), and the top filling layer is printed using 7:4:3 biomaterial ink (red). A denser supporting layer was printed between the grid and the filling layer using 9:4:3 biomaterial ink to avoid the collapse of the 7:4:3 biomaterial ink during the printing process. The inset image is viewed from the bottom up. (**D**) Grid-filled quadrilateral prism with a “HIT” pattern. The grid was printed using 9:4:3 biomaterial ink (blue), and the “HIT” pattern was printed along Z-direction using 7:4:3 biomaterial ink (red). The image on the right shows the cross-section at the position of the black dashed line.

**Table 1 bioengineering-10-01358-t001:** The values of variables for RSM.

Variables *X*_i_	Levels		
	−1	0	1
Gel (wt%)	6	8	10
SA (wt%)	3	4	5
MC (wt%)	2	3	4

**Table 2 bioengineering-10-01358-t002:** Experimental results of RSM in the viscosity properties of biomaterial inks.

	n_Prediction_	n_Reality_	n_Accuracy_	K_Prediction_	K_Reality_	K_Accuracy_
10:4:3	0.45	0.44	97.06%	161.44	146.02	90.44%
6:5:4	0.55	0.55	99.47%	81.57	80.98	99.28%
7:4:4	0.59	0.58	98.59%	62.85	60.52	96.30%
7.5:3:2	0.67	0.65	96.85%	12.80	15.41	79.62%
9:5:3	0.46	0.46	99.32%	184.07	147.79	80.29%

**Table 3 bioengineering-10-01358-t003:** Optimization results of filament size.

	Printing Speed (mm/s)	Length of Filament (Design Values: 15 mm)	Accuracy
Unoptimized	8	14.12 ± 0.05	94.20%
	12	14.07 ± 0.03	93.81%
	16	13.98 ± 0.07	93.23%
Optimized (*t*_0_ = 0.1 s)	8	14.92 ± 0.02	99.47%
	12	14.86 ± 0.01	99.13%
	16	14.98 ± 0.03	99.89%

## Data Availability

Raw data is available on request.

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
