# Peer review of "Investigation of Biomaterial Ink Viscosity Properties and Optimization of the Printing Process Based on Pattern Path Planning"

_bioengineering, 2023, doi:10.3390/bioengineering10121358_

Round 1

Reviewer 1 Report

Comments and Suggestions for Authors
  • The manuscript addresses the lack of discussion on the controllable preparation and printability of bioinks with different viscosities, providing insights into the adjustment of the rheological properties of bioinks. The use of response surface methodology (RSM) to establish the relationship between material components and rheological properties of bioinks is commendable. However, the paper lacks a discussion on the limitations of preparing and printing bioinks with different viscosities, which could have provided a more comprehensive understanding of the topic. Additionally, the effect of crosslinking and optimization on the shrinking of the printed scaffolds needs to be discussed in more detail.

Reviewer 2 Report

Comments and Suggestions for Authors

Nice, detailed work on defining bioinks that will help a lot of other researchers.  The authors should cite the original/first papers where the terms bionic were used.

Comments on the Quality of English Language

some errors: researches (line 53), bondings (line 75),  stopped reviewing for editing after that, but authors need to review for grammar thoroughly.

Reviewer 3 Report

Comments and Suggestions for Authors

The authors have conducted a thorough investigation into the influence of bioink rheological properties on its printability, incorporating a model to predict the bioink’s viscosity characteristics. The validation of this model was achieved through statistical methods, demonstrating its high significance. Comprehensive characterizations were performed to optimize the printing results, culminating in a manuscript of excellent quality. In order to enhance clarity and fortify the manuscript’s structure, addressing the following inquiries is recommended prior to publication:

  1. The choice of measuring the compression modulus as opposed to other mechanical properties necessitates further clarification. What renders the compression modulus a critical parameter for evaluating bioink, and how does it surpass other mechanical properties in importance within this context?
  2. Regarding the bioink composition, clarification is required on whether alterations in the ratios of the three distinct components lead to changes in the concentration of each individual component, or if such changes are exclusive to sodium alginate. A comprehensive breakdown of the component concentrations for each bioink formulation would contribute significantly to the manuscript’s transparency and comprehensibility.
